# Characterization of Rickets Type II Model Rats to Reveal Functions of Vitamin D and Vitamin D Receptor

**DOI:** 10.3390/biom13111666

**Published:** 2023-11-19

**Authors:** Yuichiro Iwai, Ayano Iijima, Satoko Kise, Chika Nagao, Yuto Senda, Kana Yabu, Hiroki Mano, Miyu Nishikawa, Shinichi Ikushiro, Kaori Yasuda, Toshiyuki Sakaki

**Affiliations:** 1Department of Pharmaceutical Engineering, Faculty of Engineering, Toyama Prefectural University, 5180 Kurokawa, Imizu 939-0398, Toyama, Japan; u256004@st.pu-toyama.ac.jp (Y.I.); ayano.iijima1@persol.co.jp (A.I.); skise1998@stu.kanazawa-u.ac.jp (S.K.); nagao.chika@ma.medience.co.jp (C.N.); u018014@st.pu-toyama.ac.jp (Y.S.); u018033@st.pu-toyama.ac.jp (K.Y.); h-mano@nippongene.com (H.M.); 2Department of Biotechnology, Faculty of Engineering, Toyama Prefectural University, 5180 Kurokawa, Imizu 939-0398, Toyama, Japan; m-nishikawa@pu-toyama.ac.jp (M.N.); ikushiro@pu-toyama.ac.jp (S.I.)

**Keywords:** vitamin D, vitamin D receptor, genome editing, rickets, alopecia

## Abstract

Vitamin D has been known to exert a wide range of physiological effects, including calcemic, osteogenic, anticancer, and immune responses. We previously generated genetically modified (GM) rats and performed a comparative analysis of their physiological properties to elucidate the roles of vitamin D and vitamin D receptor (VDR). In this study, our primary goal was to investigate the manifestations of type II rickets in rats with the VDR(H301Q) mutation, analogous to the human VDR(H305Q). Additionally, we created a double-mutant rat with the VDR(R270L/H301Q) mutation, resulting in almost no affinity for 1,25-dihydroxy-vitamin D3 (1,25D3) or 25-hydroxy-vitamin D3 (25D3). Notably, the plasma calcium concentration in *Vdr*(R270L/H301Q) rats was significantly lower than in wild-type (WT) rats. Meanwhile, *Vdr*(H301Q) rats had calcium concentrations falling between those of *Vdr*(R270L/H301Q) and WT rats. GM rats exhibited markedly elevated plasma parathyroid hormone and 1,25D3 levels compared to those of WT rats. An analysis of bone mineral density in the cortical bone of the femur in both GM rats revealed significantly lower values than in WT rats. Conversely, the bone mineral density in the trabecular bone was notably higher, indicating abnormal bone formation. This abnormal bone formation was more pronounced in *Vdr*(R270L/H301Q) rats than in *Vdr*(H301Q) rats, highlighting the critical role of the VDR-dependent function of 1,25D3 in bone formation. In contrast, neither *Vdr*(H301Q) nor *Vdr*(R270L/H301Q) rats exhibited symptoms of alopecia or cyst formation in the skin, which were observed in the *Vdr*-KO rats. These findings strongly suggest that unliganded VDR is crucial for maintaining the hair cycle and normal skin. Our GM rats hold significant promise for comprehensive analyses of vitamin D and VDR functions in future research.

## 1. Introduction

Vitamin D exerts a wide range of physiological effects, including calcemic, osteogenic, anticancer, and immune responses. The active form of vitamin D3, 1,25D3, binds to the vitamin D receptor (VDR) to form a heterodimer with RXR, subsequently binding to the vitamin D response element (VDRE) and regulating the expression of various genes. A mutation causing a loss of function in the VDR gene results in the inactivity of VDR-dependent vitamin D actions, leading to symptoms like bow legs/knock knees due to osteodystrophy and alopecia due to hair follicle dysplasia, known as vitamin D-dependent type II rickets (VDDRII) [1,2,3,4]. Bone dysgenesis can be ameliorated through symptomatic treatment with high calcium intake; however, no appropriate treatment has been developed for alopecia. Recent studies have suggested multiple effects of vitamin D and/or VDR, including the VDR-independent effects of 1,25D3, the VDR-dependent or VDR-independent effects of 25D3, and the ligand-independent effects of the VDR [5,6,7]. We generated genetically modified (GM) rats using genome editing to distinguish between these effects (Table 1) [8,9,10]. The findings are as follows: (1) A comparison of WT and *Vdr*(R270L) rats highlights the action of VDR-dependent 1,25D3, and (2) a comparison of *Vdr*(R270L) and *Cyp27b1*-KO rats emphasizes the action of VDR-independent 1,25D3. (3) A comparison of *Vdr*(R270L/H301Q) and *Vdr*-KO rats reveals an unliganded VDR action, and (4) a comparison of *Vdr*(R270L) and *Vdr*(R270L/H301Q) rats demonstrates a VDR-dependent effect of 25D3. Recently, we unveiled that a VDR-expressing adenoviral vector (AdV) is a promising tool for elucidating the function of a VDR [10]. In particular, the expression system for mutated VDRs using an AdV in HaCaT-VDR-KO cells represents a unique strategy. The affinities of various VDRs for 1,25D3 and 25D3 were measured using our original in vitro system [10,11]. Using these systems, the nuclear translocation of VDR(H301Q) was induced in HaCaT-VDR-KO cells in the presence of 10 nM 1,25D3 and partially induced in the presence of 100 nM 25D3. In contrast, no nuclear translocation of VDR(R270L/H301Q) was observed under the same conditions (Table 2). We hypothesized that VDR(H301Q) and VDR(R270L/H301Q) have nearly the same affinities for 25D3. However, a clear difference in ligand affinity was observed between VDR (H301Q) and VDR (R270L/H301Q) in the in vitro system (Table 2). These results suggest that the additional R270L mutation in VDR(H301Q) affects hydrogen formation with the 1α-OH group and the 25-OH group. Notably, VDR, with almost no affinity for either 1,25D3 or 25D3, revealed the mechanism of the unliganded action of VDR. As shown in our previous study, *Vdr*(R270L/H301Q) rats showed no alopecia, whereas *Vdr*-KO rats exhibited alopecia [10]. These results strongly suggest that unliganded VDR is essential for maintaining the hair cycle. In this study, we focused on a comparison of WT, *Vdr*(H301Q), and *Vdr*(R270L/H301Q) rats in terms of their plasma Ca, PTH, and 1,25D3 levels and bone structures. The unliganded VDR function is discussed based on the remarkable differences in hair cycling and skin senescence between *Vdr*(R270L/H301Q) and *Vdr*-KO rats. Our GM rat series, including *Vdr*(H301Q) and *Vdr*(R270L/H301Q) rats, may have strong implications for elucidating the molecular mechanisms of vitamin D and VDR action (Table 1).

## 2. Materials and Methods

### 2.1. Materials

25D3 and 1,25D3 were purchased from Cayman Chemical Company (Ann Arbor, MI, USA). The other chemicals were commercially available and of the highest quality.

### 2.2. Animals and Diets

Jcl:Wistar rats were purchased from Nihon Clea Co., Ltd. (Tokyo, Japan). The *Vdr*(H301Q) and *Vdr*(R270L/H301Q) rats were generated using the CRISPR-Cas9 genome editing system as previously described [10]. Embryo microinjection was performed by KAC Co., Ltd. (Kyoto, Japan). The resulting heterozygotes were crossed with each other to obtain homozygotes and wild-type rats. All rats were housed in a room controlled at room temperature (22–26 °C), 50–55% humidity, and a 12 h light/dark cycle. They were given food and water ad libitum and fed a CE-2 formula diet (Crea Japan Co., Ltd., Tokyo, Japan) containing 1.15% Ca and 2750 IU vitamin D3/kg feed [8]. All animal experiments were conducted in accordance with the Toyama Prefectural University Animal Experiment Guidelines after receiving approval from the Toyama Prefectural University Animal Research Ethics Committee.

### 2.3. HE Staining

Skin samples were prepared from WT (15 weeks), *Vdr*(H301Q) (15 weeks), *Vdr*(R270L/H301Q) (15 weeks), and *Vdr*-KO (14 weeks) male rats, treated with 4% PFA (FUJIFILM Wako Pure Chemical Corporation, Osaka, Japan), and fixed at 4 °C for 15 h. The resultant samples were then filled with O.T.C compound (Sakura Finetek, Japan), frozen in liquid nitrogen, and then cut to a thickness of 20–25 μm using a cryostat microtome (Leica, Tokyo, Japan). The resulting cryosections were mounted on glass slides. The 4% PFA solution was dropped onto the sample glass, incubated at room temperature for 10 min, and then washed with water for 10 min. Hematoxylin was dropped onto the sample glass, incubated at room temperature for 15 min, and then washed with water for 10 min. Eosin–alcohol was dropped onto the sample glass, incubated for 2 min at room temperature, and washed with water for 2 min, 70% EtOH for 2 min, 80% EtOH for 2 min, 90% EtOH for 2 min, and 100% ethanol. An additional 2 min wash with EtOH was followed by two washes with xylene. The obtained HE-stained samples were observed using a phase contrast microscope (Olympus, Tokyo, Japan) (Figure 1).

### 2.4. Preparation of Primary Culture Cells 

Back skin was excised from 15-week-old WT, *Vdr*(H301Q), and *Vdr*(R270L/H301Q) male rats, and white adipose tissue was removed. The resultant skin samples were incubated for 18 h at 4 °C in 14 mL of CnT-07 epithelial growth medium (CELLnTEC advanced cell systems AG, Bern, Switzerland) containing 5 mg/mL collagenase and washed with PBS to remove excess collagenase. Next, the resulting skin samples were incubated in 5 mL of Tryp Express (Thermo Fisher, Waltham, MA, USA, #12604013) for 15 min at 37 °C, and then 25 mL of CnT-07 medium was added, and the samples were vigorously pipetted to disassociate cell–cell adhesions. The resulting cell solution was centrifuged at 2000 rpm for 5 min to obtain a pellet containing keratinocyte cells. Next, the pellet was suspended in 4 mL of CnT-07 medium containing penicillin–streptomycin and amphotericin B at final concentrations of 100 units/mL and 0.5 μg/mL, respectively, and the collected cells were cultured in a 6 cm collagen-coated dish. After 2 days of culture, the medium was changed from CnT-07 medium to Keratinocyte Growth Medium 2 (PromoCell, Heidelberg, Germany).

### 2.5. Immunofluorescence Staining 

Primary culture cells prepared from the dorsal skin of the WT, *Vdr*(H301Q), or *Vdr*(R270L/H301Q) male rats were cultured in 6 cm collagen-coated dishes (Sumitomo Bakelite Co., Ltd., Tokyo, Japan). After washing the cells with PBS, a CnT-07 medium containing 10 nM of 1,25D3 was added to observe the ligand-dependent nuclear translocation of VDR. At 120 min after the 1,25D3 addition, the cells were fixed with 4% PFA (Fujifilm Wako Pure Chemical Industries, Ltd., Tokyo, Japan) for 15 min at 4 °C. PBS containing 0.01% Triton-X was used for cell permeabilization, and 0.1% BSA/PBS was used for blocking to prevent non-specific detection. For the immunofluorescence detection of VDR, the Anti-VDR antibody (D2K6W) Rabbit mAb (Cell Signaling Technology, Danvers, MA, USA) and Alexa Flour 488 goat anti-rabbit IgG (Invitrogen, Carlsbad, MA, USA) were used as the first and secondary antibodies, respectively [10]. After staining the nuclei with DAPI, VDR was visualized using the Invitrogen EVOS FL Cell Imaging System (Thermo Fisher Scientific, Waltham, MA, USA).

### 2.6. Western Blot Analysis

The back skin was excised and homogenized using a Minilis personal homogenizer (ber-tin technology, Montigny-le-Bretonneux, France). The resulting tissue lysate containing 10 mg of protein was applied to each lane of the gel and subjected to SDS-PAGE on a 4–20% linear gradient polyacrylamide/SDS gel. After electrophoresis, the proteins in the gel were electrotransferred onto a PVDF membrane. The PVDF membrane was then incubated in Tris-buffered saline with 0.05% Tween 20 (TBS-T) containing 5% skim milk, followed by anti-VDR antibody (D2K6, rabbit mAb) (Cell Signaling Technology, Danvers, MA, USA) or anti-β-Actin Rabbit Ab (Cell Signaling Technology, Danvers, MA, USA) [10]. The PVDF membrane was washed three times with TBS-T and incubated with horseradish-peroxidase-conjugated goat anti-rabbit IgG (Cell Signaling Technology, Danvers, MA, USA). The PVDF membrane was washed with TBS-T, followed by enhanced chemiluminescence immunodetection (Amersham Pharmacia Biotech, Buckinghamshire, UK).

### 2.7. Computed Tomography 

A computed tomography (CT) analysis of the left femur was performed to examine the morphological characteristics and bone mineral density (BMD) of the rat bones [8]. The femur was scanned using X-ray CT (Latheta LCT-200; Hitachi Aloka Medical, Tokyo, Japan) with a voltage of 50 kVp, a current of 500 μA, an integration time of 3.6 ms, an axial field of view of 48 mm, an an isotropic voxel size of 48 μm. The BMD was calculated using LaTheta software (Hitachi Aloka Medical). A threshold density of 160 mg/cm^3^ was chosen to distinguish between calcified and non-calcified tissues. Then, 3D images of the femur were constructed from the scanned images using VGSTUDIO 3.2 software (Volume Graphics, Heidelberg, Germany).

### 2.8. Measurement of Calcium Metabolism Parameters in Plasma

The plasma Ca concentration was measured using a Calcium E-Test Wako (FU-JIFILM Wako Wako Pure Chemical, Osaka, Japan). The plasma PTH concentration was measured using a Rat Intact PTH ELISA Kit (Immutopics Inc., San Clemente, CA, USA). The plasma 1,25D3 concentration was measured using a 1,25-D3 ELISA Kit (Immundiagnostik, Bensheim, Germany) [8]. 

### 2.9. Statistical Analysis

The statistical significance of differences in Figures and Tables was analyzed using Student’s *t*-test. The criterion for significance was *p* < 0.05.

## 3. Results

### 3.1. HE Staining of Dorsal Skin

The appearance of WT, mutant *Vdr*(H301Q), *Vdr*(R270L/H301Q), and *Vdr*-KO phenotypes in rats fed a CE-2 diet containing 1.15% Ca at 15 weeks after birth was documented in our previous studies [8,10]. In *Vdr*-KO rats, hair loss progressed over time, and the elasticity and softness of the skin markedly decreased, resulting in the wavy appearance of their skin (Appendix A) [8]. In contrast, the *Vdr*(H301Q) and *Vdr*(R270L/H301Q) rats did not show alopecia [10]. The HE staining of the dorsal skin of the *Vdr*(H301Q) and *Vdr*(R270L/H301Q) rats revealed normal hair follicles and skin maintenance without dermal cyst formation (Figure 1). On the other hand, a *Vdr*-KO rat showed remarkable dermal cyst formation (Figure 1D). 

### 3.2. Western Blot Analysis

The expression of VDR(H301Q) or VDR(R270L/H301Q) in the back skin of the corresponding rats was examined using a Western blot analysis. Prominent VDR expression was observed in both GM rats (Appendix A). The levels of mutant VDRs were not significantly different from VDR levels in WT rats. The apparent molecular weight of the mutant VDRs was approximately 58 kDa, the same as that of the WT. These results indicated that the VDR(H301Q) and VDR(R270L/H301Q) proteins were appropriately expressed in the corresponding GM rats. Additionally, 10 μg of protein was applied to each lane for the detection of VDRs and β-actin, respectively. M: molecular size marker proteins (blue pre-stained protein Standard, Broad Range P7718S, New England BioLabs. Inc., Ipswich, MA, USA).

### 3.3. Nuclear Translocation of VDR in the Presence of 10 nM of 1,25D3

Primary keratinocytes were prepared from WT, *Vdr*(H301Q), or *Vdr*(R270L/H301Q) rats. The remarkable nuclear translocation of VDR was observed in WT and *Vdr*(H301Q) upon adding 10 nM of 1,25D3 (Figure 2). In contrast, no nuclear translocation was observed in *Vdr* (R270L/H301Q) under similar conditions. These results are consistent with our previous study [10], which showed almost no affinity of VDR(R270L/H301Q) for 1,25D3. 

### 3.4. Plasma Ca, PTH, and 1,25D3 Levels in WT and GM Rats

The plasma Ca level was decreased in the order of WT, *Vdr*(H301Q), and *Vdr*(R270L/H301Q) rats at 10, 12, and 15 weeks of age. The Ca levels in both GM rats were significantly reduced compared with the WT rats (Figure 3, Table 3). In contrast, the level of parathyroid hormone (PTH), whose secretion is induced by the reduced plasma Ca level via calcium-sensing receptor (CaSR) in the parathyroid, was significantly increased in both GM rats. Similarly, 1,25D3 levels in *Vdr*(H301Q) and *Vdr*(R270L/H301Q) rats were significantly higher than in WT rats (Figure 3 and Table 3). These results are quite consistent with the fact that PTH induces the expression of the Cyp27b1 gene [8].

### 3.5. Bone Formation and Bone Mineral Density in WT and GM Rats 

A μCT analysis revealed abnormal morphological changes in the cortical and trabecular bones of the femurs of the *Vdr*(H301Q) and *Vdr*(R270L/H301Q) rats (Figure 4). As shown in Table 4, the bone volumes of the *Vdr*(H301Q) and *Vdr*(R270L/H301Q) rats were not significantly different from those of the WT rats. However, the bone length of the GM rats was significantly shorter than that of the WT rats. In contrast, the maximum bone width of the GM rats was significantly shorter than that of the WT rats. The bones of the *Vdr*(R270L/H301Q) rats were morphologically more abnormal compared to those of the *Vdr*(H301Q) rats. It is noted that these morphologic features were also observed in *Vdr*(R270L), *Vdr*-KO, and Cyp27b1-KO rats, whereby the rupture of the epiphyseal cartilage growth plate and the disorganization of cartilaginous growth plates were observed [8]. The disorganized cartilaginous growth plates likely caused thick and short bones in the GM rats.

In the WT rat femurs, the cortical BMD was higher at the diaphysis than at the epiphysis, resulting in a gentle, mountain-shaped BMD distribution, as shown in Figure 4A. In contrast, both the *Vdr*(H301Q) and *Vdr*(R270L/H301Q) rats showed decreased cortical BMD among all sections in the femur, and the latter showed more reduced BMD than the former. 

The femur trabecular bone was highly distributed in the epiphysis in the WT rats, which showed a U-shaped curve when the trabecular BMD was plotted from proximal to distal along the major axis of the femur (Figure 4B). In contrast, the trabecular distribution in the *Vdr*(H301Q) and *Vdr*(R270L/H301Q) rats was substantially different from that in the WT rats, which showed a flat curve (Figure 4B). Both *Vdr*(H301Q) and *Vdr*(R270L/H301Q) rats showed increased trabecular BMD in all sections in the femur, with the latter showing a higher BMD than the former. Figure 4C shows the BMD values of the cortical and trabecular bones. The BMD of the cortical bone in the rats decreased in the order WT, *Vdr*(H301Q), and *Vdr*(R270L/H301Q), whereas the BMD of the trabecular bone increased in the same order.

## 4. Discussion

### 4.1. Hair and Skin

We successfully generated rats expressing VDR(R270L/H301Q), which exhibited almost no binding affinity for 1,25D3 and 25D3. Two amino acid substitutions in the ligand-binding pocket of VDR nearly completely abolished its ligand binding ability [10,11]. A Western blot analysis indicated the expression of the VDR(R270L/H301Q) protein with a molecular mass of 58 kDa. The expression level of VDR(R270L/H301Q) was comparable to that of WT-VDR. These results imply that VDR(R270L/H301Q) has a nearly identical tertiary structure and stability to WT-VDR. Consequently, the *Vdr*(R270L/H301Q) rat serves as an optimal model animal for analyzing the unliganded VDR function. As described in the Introduction, (3) comparing *Vdr*(R270L/H301Q) with *Vdr*-KO rats, we can observe unliganded effects of VDR. The *Vdr*-KO rats developed alopecia, whereas the *Vdr*(R270L/H301Q) rats did not. These results strongly support the idea that unliganded VDR contributes to the maintenance of the hair cycle. Although the molecular mechanism of hair loss in *Vdr*-KO rats remains unknown, it is possible that the complex formation of VDR with both RXR and Hairless (Hr) may be involved in the activation of Wnt signaling by interacting with β-catenin and Lef1 [12,13,14,15] in keratinocytes [16]. This suggests that VDR is essential for anagen re-entry. The formation of a complex between unliganded VDR and RXR appears to be particularly crucial for maintaining the hair cycle, as indicated by human VDR(V346M), which loses the ability to interact with RXR, thus causing alopecia. These findings imply that unliganded VDR(R270L/H301Q) forms a complex with RXR, and interacts with Hr. Teichert et al. [17] demonstrated that VDR directly or indirectly regulates the expression of genes required for hair follicle cycling, including Hedgehog (Hh) signaling. In contrast, Joko et al. [18] reported that VDR is a novel regulator of the catagen phase. They demonstrated that VDR is an essential regulator of hair follicle regression through the progression of programmed cell death in conditional knockout (cKO) mice generated using keratin-14 promoter-driven Cre recombinase. Surviving epithelial strands, due to low apoptosis, appear to cause hair follicle disruption and dermal cyst formation. Hence, hair loss is closely associated with dermal cyst formation. *Vdr*(R270L/H301Q) rats did not exhibit hair loss or dermal cyst formation. These results suggest that unliganded VDR in keratinocytes plays a pivotal role in regulating hair follicle regression, thereby inducing the transition from the catagen to the telogen phase of the hair follicle by promoting apoptosis of the epithelial strand. VDR is expressed in the nuclei of normal epithelial strand-forming cells during the catagen phase, implying that it promotes cell death in epithelial strands through transcriptional regulation. This speculation, combined with our results, leads to the hypothesis that unliganded VDR promotes the death of epithelial strands through transcriptional regulation in the nuclei. We intend to confirm this in future studies. Recently, a novel model of hair follicle development was reported [19]; however, the role of VDR in this context remains unclear.

### 4.2. Bone Formation

As shown in Table 1, comparing the *Vdr*(H301Q) and *Vdr*(R270L/H301Q) rats revealed the cumulative VDR-dependent effects of 1,25D3 and 25D3. The plasma calcium level in the *Vdr*(R270L/H301Q) rats was lower than that in the *Vdr*(H301Q) rats. In contrast, the plasma PTH and 1,25D3 levels in the Vdr(R270L/H301Q) rats were higher than those in the *Vdr*(H301Q) rats (Figure 4). Bone formation in the *Vdr*(R270L/H301Q) rats was more aberrant than in the *Vdr*(H301Q) rats. These findings indicate that type II rickets symptoms in the *Vdr*(R270L/H301Q) rats were more severe, likely due to the lower plasma Ca levels. 

Comparing the *Vdr*(R270L) [8] and *Vdr*(R270L/H301Q) rats revealed the VDR-dependent effects of 25D3. However, no significant differences were observed in the plasma Ca levels and bone mineral density of the cortical and trabecular bones between the *Vdr*(R270L) [8,9] and *Vdr*(R270L/H301Q) rats in this study. This suggests that the VDR-dependent effects of 25D3 are non-essential for maintaining plasma Ca levels and bone formation. Therefore, the observed difference between the *Vdr*(H301Q) and *Vdr*(R270L/H301Q) rats appears to be attributed to variations in plasma Ca levels due to the VDR-dependent effects of 1,25D3. Table 1 summarizes the relationship between GM rats and vitamin D/VDR function. Low plasma Ca concentrations and abnormal bone formation were observed in all GM rats. These results indicate that VDR-dependent 1,25D3 action is vital for maintaining plasma calcium levels and normal bone formation. However, among the GM rats, the Cyp27b1-KO rats exhibited the lowest plasma Ca levels, the most abnormal bone formation, and the smallest body size [8]. These results suggest that not only the VDR-dependent effects of 1,25D3 but also the VDR-independent effects of 1,25D3 contribute to the maintenance of plasma calcium levels and normal bone formation. Alopecia and abnormal skin formation were observed only in *Vdr*-KO rats. These results strongly support the idea that unliganded VDR is essential for the regulation of hair follicle cycling and skin homeostasis. A proteome analysis of back skin proteins of the WT and *Vdr*-KO rats at 28 weeks of age revealed a significant decrease in various proteins related to oxidative phosphorylation, including NADH dehydrogenase, cytochrome c oxidase, and ATP synthetase in *Vdr*-KO rats. These findings suggest that enhanced skin aging in *Vdr*-KO rats is closely associated with mitochondrial dysfunction. Multi-omix analyses are expected to unveil the roles of vitamin D and VDR in various organs and tissues. 

It is known that 1,25D3 induces rapid Ca influx into cells via VDR [6,20] and/or 1,25D3MARRS (ERp57) in the plasma membrane [21]. If the effect of VDR present in caveolae is important in rapid Ca influx into cells due to the addition of an active form of vitamin D, the magnitude of a change in the intracellular Ca concentration seems to be in the order of WT, *Vdr*(H301Q), and *Vdr*(R270L/H301Q) rats. However, if the effect of 1,25D3MARRS (ERp57) is important rather than VDR, it is possible that there is not much difference among the three groups. Thus, our GM rats are likely to become increasingly important for comprehensive analyses of vitamin D and VDR functions.

## 5. Conclusions

The unliganded VDR in keratinocytes is essential for the regulation of hair cycling and skin homeostasis. On the other hand, VDR-dependent 1,25D3 action is vital for maintaining plasma calcium levels and normal bone formation.

## Figures and Tables

**Figure 1 biomolecules-13-01666-f001:**
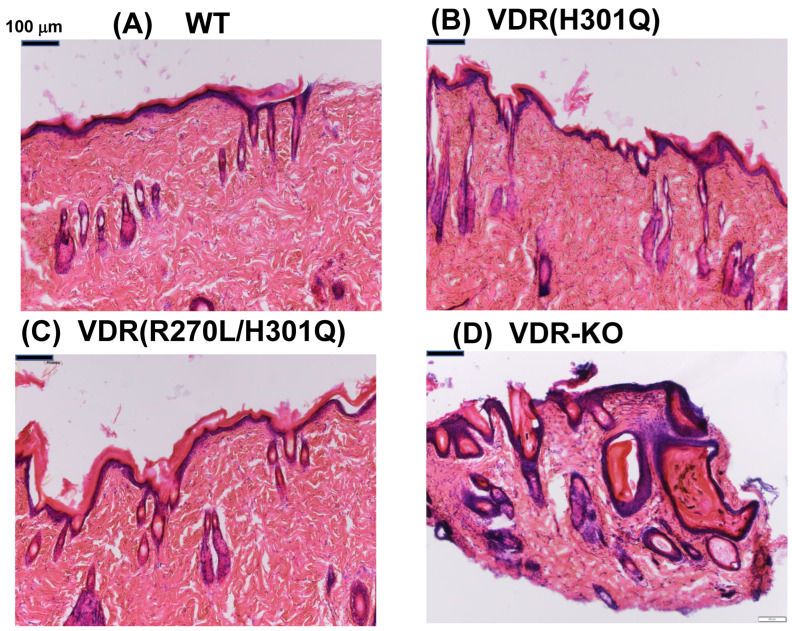
H&E staining of the dorsal skin of WT (**A**), *Vdr*(H301Q) (**B**), and *Vdr*(R270L/H301Q) (**C**), and *Vdr*-KO rats (**D**).

**Figure 2 biomolecules-13-01666-f002:**
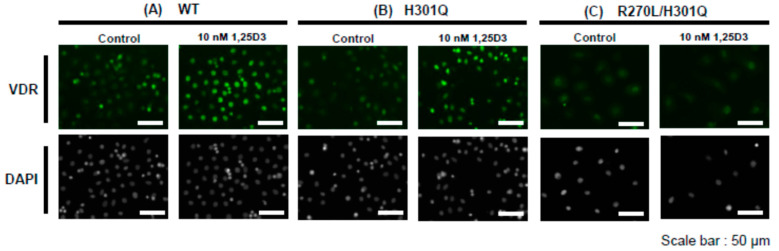
Nuclear translocation of WT-VDR, VDR(H301Q), and VDR(R270L/H301Q) by adding 10 nM of 1,25D3 in the keratinocyte primary cells prepared from WT, *Vdr*(H301Q), and *Vdr*(R270L/H301Q) rats, respectively. Control means no addition of 1,25D3.

**Figure 3 biomolecules-13-01666-f003:**
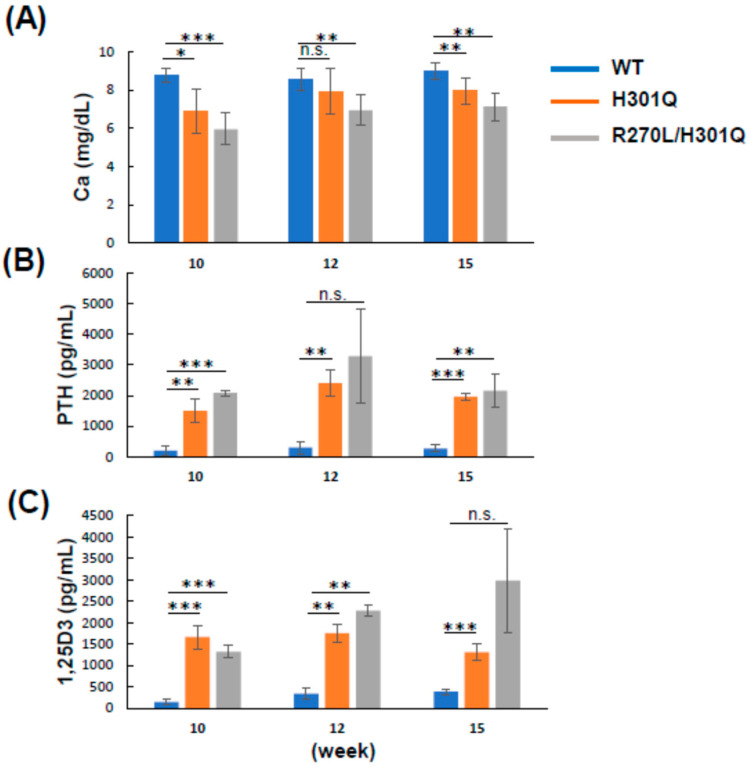
Plasma concentrations of Ca (**A**), PTH (**B**) and 1,25D3 (**C**) in WT, *Vdr*(H301Q), or *Vdr*(R270L/H301Q) rats. The values are shown as the mean ± SD (n = 4–5, n = 2–4 and n = 2–4 animals/group for Ca, PTH and 1,25D3, respectively). *: *p* < 0.05, **: *p* < 0.01, ***: *p* < 0.001, and n.s.: not significant are determined by Student’s *t*-test.

**Figure 4 biomolecules-13-01666-f004:**
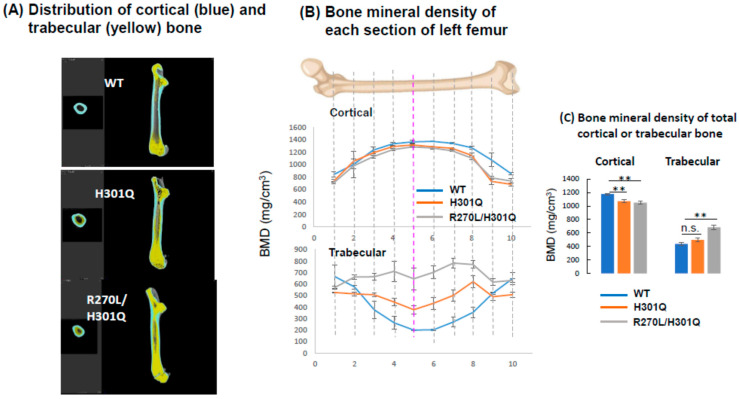
Three-dimensional deconvolution μ-CT images of femur vertical sections of WT, *Vdr*(H301Q), and *Vdr*(R270L/H301Q) rats at 15 weeks old. Cortical and trabecular bones are colored with cyan and yellow, respectively (**A**). BMD distribution between 10 horizonal sections of whole femurs of WT, *Vdr*(H301Q), and a *Vdr*(R270L/H301Q) rats. Sections 1 and 10 are the most proximal and distal sections, respectively (**B**). The bone mineral density (BMD) of the total cortical and trabecular bones are shown in (**C**). BMD values are the means ± SD (n = 4 for *Vdr*(H301Q) and *Vdr*(R270L/H301Q) rats, and n = 2 for WT rats). **: *p* < 0.01, and n.s.: not significant are determined by Student’s *t*-test.

**Table 1 biomolecules-13-01666-t001:** VDR-dependent or VDR–independent 1,25D3 or 25D3 actions and unliganded VDR function in WT and GM rats.

Rat Strain	VDR-Dependent 1,25D3	VDR-Independent 1,25D3	VDR-Dependent 25D3	VDR-Independent 25D3	Unliganded VDR
WT	◯	◯	◯	◯	◯
*Vdr*(R270L)	×	◯ *	◯	◯	◯
*Cyp27b1*-KO	×	×	◯	◯	◯
*Vdr*(H301Q)	△	◯ *	△	◯	◯
*Vdr*(R270L/H301Q)	×	◯ *	×	◯	◯
*Vdr*-KO	×	◯ *	×	◯	×

* Because of a higher plasma concentration of 1,25D3 than in the WT, the VDR-independent function of 1,25D3 in these GM rats might be enhanced compared with WT rats, although there is no evidence. The symbol ◯ means nearly the same as WT, and the symbol × means almost deficiency. The symbol △ means an intermediate which is markedly different from ◯ and ×.

**Table 2 biomolecules-13-01666-t002:** Comparison of WT and mutant VDRs with respect to their ligand binding affinity, ligand-dependent nuclear transfer, and induction of *Cyp24a1* gene expression demonstrated in our previous studies [10].

VDR	Relative Affinity for 1,25D3	Relative Affinity for 25D3	Nuclear Transfer 10 nM 1,25D3	Nuclear Transfer 100 nM 25D3	Induction of *Cyp24a1* 10 nM 1,25D3
WT	100	100	Yes	Yes	Yes
H301Q	0.2	0.2	Yes	partially Yes	Yes
R270L/H301Q	<0.001	<0.1	No	No	No

**Table 3 biomolecules-13-01666-t003:** Comparison of WT and the GM rats on plasma Ca, PTH, and 1,25D3 concentrations at 12 weeks after birth.

Rat Strain.	Ca (mg/dL)	PTH (pg/mL)	1,25D3 (pg/mL)
WT	8.56 ± 0.57	301.5 ± 191.3	344.0 ± 130.0
*Vdr*(H301Q)	7.93 ± 1.17 n.s.	2397.0 ± 419.5 **	1750.7 ± 210.9 **
*Vdr*(R270L/H301Q)	6.93 ± 0.79 **	3282.5 ± 1539.1 n.s.	2272.7 ± 127.2 **

**: *p* < 0.01, n.s.: not significant.

**Table 4 biomolecules-13-01666-t004:** Comparison of WT and the GM rats on bone morphology and BMD at 15 weeks after birth.

Rat Strain	Femur Length (cm)	Femur Max Width (cm)	Bone Volume (cm^3^)	BMD Cortical (mg/cm^3^)	BMD Trabecular (mg/cm^3^)
WT	3.9 ± 1.8	7.9 ± 0.10	0.55 ± 0.06	1178.6 ± 11.0	438.5 ± 19.0
*Vdr*(H301Q)	3.3 ± 0.8 **	8.3 ± 0.03 n.s.	0.47 ± 0.03 n.s.	1071.4 ± 19.4 **	496.9 ± 24.2 n.s.
*Vdr*(R270L/H301Q)	3.2 ± 1.1 **	8.6 ± 0.01 *	0.54 ± 0.01 n.s.	1048.7 ± 21.5 **	683.5 ± 34.7 **

*: *p* < 0.05, **: *p* < 0.01, and n.s.: not significant.

## Data Availability

The datasets generated or analyzed during the current study are available from the corresponding author (T.S.) upon reasonable request. A genomic sequence of the *Vdr* gene of the *Vdr* KO rats containing the mutated position is available in the DDBJ data base at the accession number LC764592 (http://getentry.ddbj.nig.ac.jp/top-j.html, accessed on 25 October 2023).

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
