# Peer review of "Characterization of Rickets Type II Model Rats to Reveal Functions of Vitamin D and Vitamin D Receptor"

_biomolecules, 2023, doi:10.3390/biom13111666_

Round 1
Reviewer 1 Report
Comments and Suggestions for Authors
Major comments:
1. Figures: Figure 2 is of low quality and should be replaced by a better version. Together with Figure 3 it is only of confirmatory nature and may not be worth a plain figure. Thus, shift it to the supplements.
2. Results: The Vdr mutations should also be studied on the level of gene regulation.
3. Discussion: It is unusual to refer to Figures and Tables in the Discussion section. Please reword.
Minor comments:
1. Title: "VDR(H301Q) or VDR(R270L/H301Q)" may not need to be mentioned in the title.
2. All abbreviations need to be defined at first time use and applied then consistently (e.g. 1,25D3). This applies also to the Abstract.
3. Please have consistently a space between numbers and units.
4. Gene name abbreviations have to be in italic.
Comments on the Quality of English LanguageOverall English is OK, but minor corrections are necessary.
Author Response
Thank you very much for your valuable suggestions. The manuscript was changed according to your suggestions.

Reviewer 2 Report
Comments and Suggestions for Authors
XCVXReview-biomolecules-Iwai et al-Nov-23
This MS reports on the interesting effects of two VDR gene variants on the affinity of calcitriol and calcidiol for the VDR and on bone structure and circulating calcium, calcitriol and parathyroid hormone concentrations in rickets type II model rats, the findings suggesting that un-liganded VDR is capable of allowing normal development of skin and hair follicles, and that these mutations reduce cortical bone strength, but increase trabecular bone strength, to variable degrees. The authors further suggest that studies of the effects of these mutations could be useful for determining the independent effects of the unliganded VDR and of calcitriol in other areas of vitamin D physiology. The MS is clearly written, but the labelling of some figures could be improved.
General comments.
1.it is possible that there could be differences in the activity of the 1-alpha hydroxylase that produces calciferol intracellularly as a result of the VDR mutations and, since that would be functionally important, it would be of great interest to report on this matter in the skin [and hopefully in the bone], using standard histochemical methodology, in normal, mutant and VDR-KO animals, even if only to rule this possibility out.
2.The VDR has important functions, both in the nucleus, as studied in this work and also in cell wall caveolae where rapid non-genomic effects are induced through rapid rises in intra-cellular calcium and those functions are well understood. I wonder, therefore, whether the available data allows these separate functions to be commented upon in the animal groups studied [normal, VDR-KO and the two mutant groups]. This is because, though one might expect the VDR from the two sites, nucleus and cell wall, to show similar behaviour, this might not be the case so that being able to report on that, however briefly, would be of considerable interest.
3. Similarly, since rapid non-genomic effects of activated VDRs cause measurable changes in intracellular calcium it would be really useful to be able to provide data on that variable in the different animal groups studied if at all possible, or, if not, to comment on this as a matter, or weakness, that requires further investigation.
Specific minor comments on the text, by line number.
Line 163 uses the term ‘parameter’ when in fact this section is about variables, parameter being a specific level of a variable chosen for use as a definition, [for example for the upper and lower limits of a normal range], while variables are the actual measurements made of the item of interest in a study.
Lines 220 onwards, = the Legend to Figure 4. It was not immediately clear that the three colours were being used for the three groups of rats, since the heading naming them appears as part of the heading for the first section ‘A’, leading one to look for what each colour might be showing within each of the three groups. It would help readers, therefore, if that heading [WT, H30IQ. R270L/H301Q] could appear within the legend, with the three colours shown as small bars after each grouping, especially since such clarification is provided in Figure 5A.Review-biomolecules-Iwai et al-Nov-23
This MS reports on the interesting effects of two VDR gene variants on the affinity of calcitriol and calcidiol for the VDR and on bone structure and circulating calcium, calcitriol and parathyroid hormone concentrations in rickets type II model rats, the findings suggesting that un-liganded VDR is capable of allowing normal development of skin and hair follicles, and that these mutations reduce cortical bone strength, but increase trabecular bone strength, to variable degrees. The authors further suggest that studies of the effects of these mutations could be useful for determining the independent effects of the unliganded VDR and of calcitriol in other areas of vitamin D physiology. The MS is clearly written, but the labelling of some figures could be improved.
General comments.
1.it is possible that there could be differences in the activity of the 1-alpha hydroxylase that produces calciferol intracellularly as a result of the VDR mutations and, since that would be functionally important, it would be of great interest to report on this matter in the skin [and hopefully in the bone], using standard histochemical methodology, in normal, mutant and VDR-KO animals, even if only to rule this possibility out.
2.The VDR has important functions, both in the nucleus, as studied in this work and also in cell wall caveolae where rapid non-genomic effects are induced through rapid rises in intra-cellular calcium and those functions are well understood. I wonder, therefore, whether the available data allows these separate functions to be commented upon in the animal groups studied [normal, VDR-KO and the two mutant groups]. This is because, though one might expect the VDR from the two sites, nucleus and cell wall, to show similar behaviour, this might not be the case so that being able to report on that, however briefly, would be of considerable interest.
3. Similarly, since rapid non-genomic effects of activated VDRs cause measurable changes in intracellular calcium it would be really useful to be able to provide data on that variable in the different animal groups studied if at all possible, or, if not, to comment on this as a matter, or weakness, that requires further investigation.
Specific minor comments on the text, by line number.
Line 163 uses the term ‘parameter’ when in fact this section is about variables, parameter being a specific level of a variable chosen for use as a definition, [for example for the upper and lower limits of a normal range], while variables are the actual measurements made of the item of interest in a study.
Lines 220 onwards, = the Legend to Figure 4. It was not immediately clear that the three colours were being used for the three groups of rats, since the heading naming them appears as part of the heading for the first section ‘A’, leading one to look for what each colour might be showing within each of the three groups. It would help readers, therefore, if that heading [WT, H30IQ. R270L/H301Q] could appear within the legend, with the three colours shown as small bars after each grouping, especially since such clarification is provided in Figure 5A.
Comments on the Quality of English Languagewell written in good English with just one common misuse of a term needing correction.
Author Response

(The authors gave the same response as above.)

Round 2
Reviewer 1 Report
Comments and Suggestions for Authors
none
Comments on the Quality of English LanguageMinor edits needed